# The “White Layer Approach”: A Graftless Gingival Augmentation Technique following Vertical GBR with Occlusive Titanium Barriers

**DOI:** 10.3390/medicina59101694

**Published:** 2023-09-22

**Authors:** Fabio Perret, Erik D’Aprile, Luca De Stavola

**Affiliations:** 1Private Practice, 11020 Challand-Saint-Anselme, Italy; erik.daprile@gmail.com; 2Private Practice, Piazza Aldo Moro 7, 35030 Rubano, Italy; info@admadent.com

**Keywords:** GBR, occlusive titanium barrier, pseudo-periosteum, vertical ridge augmentation, white layer approach

## Abstract

Guided bone regeneration surgery always leads to a deformation of the soft tissues consequent to passivation of the flap. In this article, a graftless technique for the restoration of the vestibular depth and for the augmentation of adherent soft tissue, called the “white layer approach”, is proposed after a vertical GBR procedure in posterior areas. Six patients (five males and one female) with vertical bone atrophy were enrolled in the study and underwent three-dimensional bone augmentation with titanium barriers. After 6 months, during the second-stage surgery, a 0.5 mm thick layer of white pseudo-periosteum was observed underneath the titanium barrier and over the newly formed bone. The buccal portion of the pseudo-periosteum was left intentionally exposed, in order to promote the spontaneous formation of new adherent gingiva and the restoration of the original depth of the fornix. The implant insertion was then planned 3 months after the WLA in a conventional procedure. The buccal adherent soft tissue height was measured from the crestal point to the most apical point, using a periodontal probe, before the barrier removal at 3 months after the white layer approach (WLA). In all patients, a gain in adherent soft tissue varying from 5 to 8 mm was observed; the average adherent soft tissue gain (ASTG) was 6.75 mm. The vertical bone height was measured by CT scans at baseline and before the implant placement, and showed an average vertical bone gain (AVBG) of 4.08 mm. Within the limitations of this study, vertical GBR with titanium occlusive barriers (OTB) associated with the white layer approach (WLA) may represent a simplified technique for hard and soft tissue augmentation in posterior areas, even without a free gingival graft.

## 1. Introduction

Guided bone regeneration (GBR) is a well-established technique used to increase the amount of bone and provide optimal bone support for osseointegrated dental implants. The application of GBR for vertical augmentation is well documented, with high implant success rates [1,2]. As proposed by Dahlin et al., the GBR therapeutic protocol involves the surgical placement of a space-creating barrier device that protects the site of regeneration, excluding cells that may impede bone formation (epithelial cells and connective tissue) [3]. The use of PTFE non-resorbable barriers, compared to resorbable barriers, can guarantee an adequate space-making effect up to the second surgical phase, which usually takes place nine months after the first operation [2]. However, in the case of accidental exposure during the healing period, in most cases, the semipermeable PTFE barriers have to be removed within a couple of weeks, due to their contamination by oral fluids [4]. The use of occlusive titanium membranes (OTBs) has been proposed to create and maintain an isolated space [5]. An OTB is a thin, preformed titanium barrier, which, being without openings and non-permeable, can ensure better biological protection of the graft in the case of accidental exposure.

Perret et al. [6] demonstrated that the use of an intentionally exposed occlusive barrier in extraction sockets can lead to bone regeneration, despite exposure. In order to reduce the intra-surgical time of OTB adaptation, the pre-shaping of the barrier on a printed stereolithographic model (STLM) has been proposed [7,8,9]. The osteoconductive and mechanical properties, and the possibility of autoclaving the titanium barriers, can guarantee an optimal biocompatible response for this device. Furthermore, the authors of the present article observed an increase in flap thickness above the OTB during the healing period [9].

Although this phenomenon is not yet completely explained, it can be assumed that it is a consequence of the occlusivity of the barrier, which hinders the penetration of the soft tissues from the flap to the regenerative space, maintaining not only the initial thickness of the flap, but even favoring its thickening. As described by Dahlin et al. [10], a white layer of connective tissue, called the pseudo-periosteum, can be observed above the newly formed bone when a non-resorbable barrier is removed. This is a dense, connective soft tissue layer with low cellularity and no mineralization (Figure 1a,b).

A classification of this layer of the pseudo-periosteum has been proposed by Cucchi et al. [11] into three types based on morphological and histological characteristics. Some authors believe that this tissue should be maintained to protect newly formed bone and avoid bone exposure in cases of secondary healing after membrane removal [12,13]. However, further studies will be needed to fully understand the role and clinical potential of the pseudo-periosteum.

The aim of the present study is to introduce a new graftless protocol for the augmentation of adherent gingiva in posterior areas treated with vertical GBR, to be applied in the OTB removal phase.

The proposed procedure provides for the possibility of leaving a vestibular portion of the pseudo-periosteum intentionally exposed, promoting its healing by secondary intention. This procedure is performed in order to restore the depth of the vestibule in the regenerated areas and increase the amount of adherent keratinized tissue without a free gingival graft.

## 2. Materials and Methods

This case series includes 6 patients (5 female, 1 male; average 59 years old) in need of vertical bone augmentation for implant placement (Figure 2a and Figure 3a). The patients underwent 5 mandibular surgeries and 1 in the upper jaw. All the patients were systemically healthy, had never smoked, and were not taking any regular medications associated with a compromised bone healing response. Prior to treatment, informed consent was obtained from patients regarding the treatment goal and protocols. In the present article, the authors describe a new approach in order to reach the proper vestibule depth and keratinized attached tissue width by taking advantage of the secondary healing of the white layer of pseudo-periosteum.

### 2.1. Occlusive Titanium Barrier Preparation

For the selected patients, cone beam computed tomography (CBCT) (Cranex 3D, Soredex—Kavo Dental, Brea, CA, USA) was performed, and showed the reduced thickness and height of the residual bone. A lack of keratinized tissue (KT) was also evident. For each CBCT, a three-dimensional STLM (stereolithographic model) of the recipient site was printed to evaluate the case and pre-shape the titanium barrier. The GBRs were planned directly on each printed STLM. A 0.12 mm thickness OTB (Regenplate^®^ Shape 2 and Shape 4, Bio-Micron, Sweden & Martina, Due Carrare, Padova, Italy) was trimmed and shaped on each STLM to maintain a distance of 1.5 mm from the adjacent teeth, from the mental nerve, and from the mylohyoid line. The smooth side of the barrier was placed externally, while the treated rough surface was placed on the inner side of the dome, in order to obtain better blood clot adhesion [14]. Every sharp edge was bent inward with pliers to avoid damage to the flap. Various of holes were drilled with a burr (2 mm diameter) in the mesial–buccal and distal–occlusal areas of the barrier to stabilize it using fixation screws. The device was then sterilized by means of an autoclave before surgery.

### 2.2. First-Phase Surgery

After local anesthesia was performed with articaine hydrochloride 4% plus epinephrine 1:100,000 (Citocartin, Molteni Dental, Milan, Italy), a flap design was made to ensure primary tension-free closure. In detail, a crestal full-thickness incision was made with a No. 15 surgical blade, to equally divide the KT. Posteriorly, the incision ended with a vestibular oblique incision (45°) at the level of the occlusal plane. In the presence of a molar, the posterior incision was marginal and distally released. Lingually, to obtain an adequate length of the flap, a marginal horizontal incision was extended mesially to at least the two adjacent teeth, avoiding vertical incisions. To obtain passivation of the lingual flap in the middle area, the superficial fibers of the mylohyoid muscle were carefully detached by a periodontal probe. In the distal area, the full thickness of the adherent tissues of the retro-molar pad was lifted using a Prichard elevator. On the buccal side, the mesial extension of the flap involved one or two adjacent teeth and ended with a “hockey-stick” vertical incision with preservation of the papillae [15]. The full thickness of the flap was then carefully raised to locate the mental foramen area and the neurovascular bundle. A shallow periosteum incision was made from the distal to the medial vertical incision, by using a new No. 15 blade, in order to obtain flap passivation and elongation of about 8 mm. The buccal flap was further carefully “brushed” on its internal surface until the desired elongation was achieved [16]. Then, the collection of autologous bone was performed from the recipient site by using a scraper (Safescraper^®^ TWIST, META, Reggio Emilia, Italy). The bone scraping procedure was considered sufficient to activate the regional acceleratory phenomenon at the recipient site [17]. A recent systematic review shows that there is no evidence supporting creating bone perforations to promote new angiogenesis in GBR [18]. A 1:1 mixture of particulated autogenous bone and porcine-derived xenograft (Regeneross^®^, Zimmer, Columbus, OH, USA) was placed in the inner surface of the membrane.

The device was, at this point, placed on the recipient site and fixed, first on the buccal side, then on the distal–occlusal area, and finally on the lingual side using self-tapping fixation screws (0220Q-4-10; 0220Q-6-10, Cizeta, Sweden & Martina, Due Carrare, Padova, Italy) (Figure 2b). Contact between the barrier and neighboring teeth was carefully avoided, keeping a minimum distance of 1.5 mm. Flaps were then closed using two layers of sutures: horizontal mattress sutures were used in the first layer, and then single interrupted sutures were placed to ensure an adequate closure, especially on the edges of the flaps. Vertical incisions were then closed with single sutures.

### 2.3. Second-Phase Surgery

The barrier was removed 6 months after vertical augmentation, to allow a sufficient period of healing in order to obtain hard tissue regeneration. In all cases, a lack of keratinized tissue and vestibule depth was evident at the second surgical phase (Figure 2c and Figure 3b—the green arrows show the coronal shifting of the mucogingival junction). A buccal horizontal full-thickness incision was placed in the alveolar mucosa, extending to the titanium barrier surface, at the same level of the mucogingival junction of the adjacent teeth. The flap was completed with two small mesial and distal vertical releasing incisions that occlusally reached the crestal residual keratinized gingiva. This access flap allows the easy removal of the fixation screws and the titanium barrier (Figure 2d).

The absence of holes in the barrier surface, contrary to the perforated titanium meshes, hinders the penetration of connective tissue, permitting very fast and easy removal of the titanium barrier.

A thin white layer of dense connective tissue, called the pseudo-periosteum, was always observed underneath the titanium occlusive barrier. It was a type 1 pseudo-periosteum with a thickness < 1 mm according to the classification proposed by Cucchi et al. [11]. This layer was left in place to protect the new regenerated hard tissue, promoting its maturation and corticalization (Figure 2e and Figure 3c).

The alveolar mucosa apical to the incision line was sutured to the bottom of the vestibule, with 6/0 resorbable sutures anchored to the apical periosteum. The lingual flap was placed crestally in the same original position. Compression sutures were applied with the purpose of stabilizing the primary flap against the pseudo-periosteum and the newly regenerated bone.

In this way, a buccal portion of the pseudo periosteum was left intentionally exposed, with second-intention healing in order to obtain an increase in vestibulum depth and in the keratinized tissue (Figure 2f and Figure 3d).

Rinses with 0.20% chlorhexidine mouthwash were prescribed, suspending the use of the toothbrush until the sutures were removed.

At the check-up after one week, the first healing tissue was visible (Figure 2g and Figure 3e). After two weeks, the area was partially epithelized, with a good increase in thickness.

After six weeks, a band of adherent and keratinized gingiva and the mucogingival line were created, aligned with the mucogingival line of the adjacent remaining teeth.

After three months, corticalization and bone maturation were observed radiographically, as well as the restoration of the original gingival anatomy (Figure 2h and Figure 3f—the yellow arrows show the new position of the mucogingival line).

A small flap was raised, and the implant sites were prepared in a prosthetically driven manner, using dedicated drills according to the manufacturer’s instructions. The implants were then placed with a minimum torque of 35 ncm, placing a healing abutment (Figure 2i). In one case, immediate loading with a splinted temporary crown was performed.

### 2.4. Prosthetic Loading

After three months, the healing abutment was replaced with a permanent abutment, and acrylic resin provisional restorations were placed. They were left in situ for two months, before being replaced by definitive zirconia–ceramic restorations. A radiographic follow-up was performed before prosthetic finalization for all cases to verify the conditions of hard peri-implant tissues (Figure 4).

### 2.5. Data Collection

The following variables were recorded by an independent operator and are presented in Table 1.

-Vertical bone gain (VBG): For each case, this was represented by the mean value of maximum vertical regeneration achieved, on both vestibular/buccal and lingual/palatal sides, by means of CT scans, measured at baseline and before the implant placement (9 months later). The measurements were carried out by means of OsiriX software (v. 3.5.1—32 bit, Pixmeo, Geneva, Switzerland).-Adherent Soft Tissue Gain (ADSG): The procedure was performed with a periodontal probe (CPC 15, Hu-friedy) measuring the buccal adherent tissue height from the crestal point to the most apical point. The gain was evaluated by recording the difference between the two values before the barrier removal and 3 months after the white layer approach (WLA).

## 3. Results

Six patients (five females and one male) with a mean age of 59 years were enrolled in the present study. All patients followed the entire protocol, without no withdrawals.

In one case (patient 1), the healing time between the GBR procedure and the WLA was 4 months. Despite the reduced healing time, the clinical consistency was good, with no penetration of the probe.

The average Vertical Bone Gain was 4.08 mm, ranging from a minimum of 3 mm to a maximum of 6 mm.

The average Adherent Soft Tissue Gain was 6.75 mm, ranging from a minimum of 5 mm to a maximum of 8 mm.

## 4. Discussion

Guided bone regeneration surgery always leads to a deformation of the soft tissues consequent to passivation of the flap. As a result, there is often a loss in the vestibule depth and a coronal migration of the mucogingival line, resulting in the loss of keratinized tissue. In order to maintain the stability of the peri-implant crestal bone over the medium-long term, the need for adequate gingival thickness around the neck of the implant has been highlighted. Furthermore, an adequate depth of the vestibule guarantees the health of the peri-implant marginal mucosa, which should remain stable during phonation and chewing movements.

In order to restore the soft tissue anatomy following vertical augmentation in the posterior areas, some authors proposed a surgical protocol that involves the uncovering of the implants with a partial thickness flap and the simultaneous positioning of a free gingival graft on the buccal side of the regenerated site, in order to restore the correct depth of the vestibule [19]. This surgical step is generally performed after two previous surgeries, which are the first phase of GBR and the second phase of membrane removal and implant insertion. The restoration of the keratinized tissue and the vestibule depth is then reserved for a third surgical phase. However, at this point, the preparation of the receiving periosteal bed in the posterior mandibular area must be performed with a deep partial thickness flap and consequently requires an operator with great expertise, as the risk of damaging the mental nerve bundles is quite high.

For these reasons, the approach proposed in this paper suggests the possibility of exploiting the only moment in which there is the spontaneous presence of the pseudo-periosteum, during the removal of the titanium barrier, without raising a partial-thickness flap.

Histologic studies demonstrated that the pseudo-periosteum underneath the non-resorbable barriers is composed of dense connective tissue having multi-directional fiber organization, a variable degree of cellularity, little or no vascularization, and an absence of inflammatory reactions [11]. In particular, in human and animal studies, Simion et al. [20,21] showed that most coronal portions of implants placed with a one-stage GBR using non-resorbable Ti-reinforced membranes were immersed in a dense fibrous tissue over the crestal level of the regenerated bone. Histologic analysis at 9 months showed that this tissue consisted of densely packed collagen fibers with few cells and a scarcity of blood vessels. No inflammatory reactions or epithelial tissue were present [21]. Although numerous studies about the pseudo-periosteum formation consider dense PTFE membranes, a similar type of tissue has been found underneath the non-resorbable occlusive titanium barriers (Figure 1e and Figure 2c).

These histological features of dense connective tissue seem to be compatible with secondary-intention healing; Lim et al. [12] suggested that the pseudo-periosteum formation facilitates secondary-intention healing in cases of barrier exposure.

In the present article, a surgical approach to restore the depth of the vestibule and increase the attached gingiva with secondary healing of the pseudo-periosteum has been proposed. The incision technique, as described in the materials and methods section, involves a horizontal incision in the alveolar mucosa, placed buccally at the level of the mucogingival line of the adjacent teeth. The buccal–apical portion of the flap is sutured apically, leaving exposed the layer of the pseudo-periosteum newly formed underneath the OTB. This white layer undergoes secondary healing with the aim of generating new adherent tissue. It has been observed that the new mucogingival junction tends to be aligned with the MGJ of the adjacent teeth. This is probably linked to the fact that the presence of adjacent keratinized tissue favors the re-creation of thick, keratinized tissue. The thickness of the new adherent tissue seems to be influenced by the thickening of the flap following the use of an OTB, an aspect already observed by the same authors in a previous article [9].

As regards the quality of the regenerated bone with GBR with non-resorbable barriers, it has been observed that the regenerated bone underneath the barrier is quite immature and requires a maturation period of 3–4 months in which the primary flap remains in contact with the regenerated tissue, promoting its crestal vascularization, maturation, and corticalization. It is possible to deduce that the placement of implants in more mature and corticalized regenerated bone could be favorable for the maintenance of the crestal bone.

Although many authors usually remove the newly formed pseudo-periosteum, other authors think that this tissue should be maintained to protect newly formed bone and avoid bone exposure in cases of secondary healing after membrane removal [13,22].

It is not yet clear how well the pseudo-periosteum is able to preserve bone resorption if left exposed. In any case, the portion of the buccal pseudo-periosteum that remains exposed is far from the critical crestal area of the implant neck; therefore, even a potential slight reabsorption of the regenerated volume could be well tolerated and irrelevant to the final esthetic and functional outcome.

## 5. Conclusions

In cases of vertical bone regeneration in the posterior areas, this article has demonstrated the possibility of recreating the depth of the fornix and new adherent gingiva by the secondary healing of an area of pseudo-periosteum left exposed.

The advantages compared to the FGG techniques are related to the lower invasiveness, operative simplicity, lower risk of mental nerve damage, reduction in healing time and finally the better mimesis and final esthetic outcome.

This approach can also be applied in cases of accidental exposure and the consequent anticipated removal of the titanium barrier. In this case, the implant placement must also be delayed by at least 3–4 months from the barrier removal.

Due to the limitations of the present study, including the small number of patients recruited, further randomized controlled cases will be needed to confirm the predictability of the proposed approach.

## Figures and Tables

**Figure 1 medicina-59-01694-f001:**
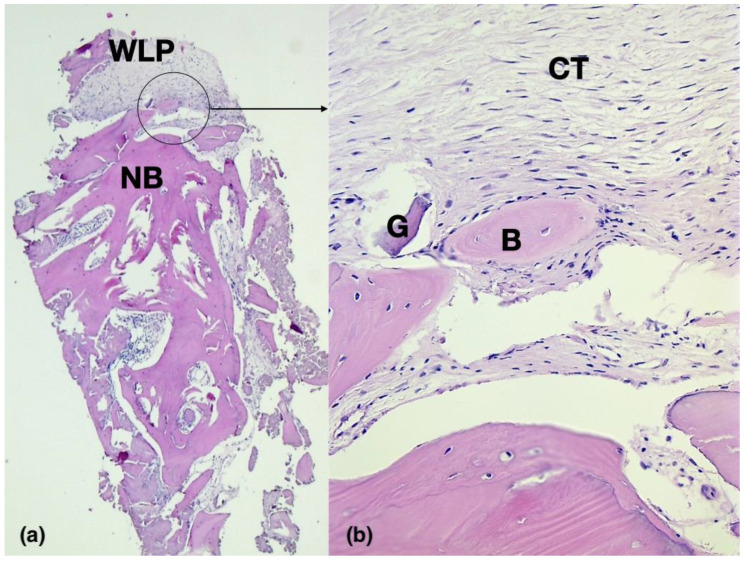
(**a**) Histologic image (25×) of a biopsy sample harvested with a trephine at the time of removal of the titanium barrier. The lower portion shows the newly formed lamellar bone (NB), while the upper portion shows the white layer of pseudo-periosteum (WLP). (**b**) Higher magnification image (100×) showing connective tissue (CT) with the presence of densely packed collagen fibers with few cells and a scarcity of blood vessels. No inflammatory reactions or epithelial tissue are present. Newly formed bone (B) portions and residual graft fragments (G) are present. Images are of H&E staining and were acquired with a digital scanner.

**Figure 2 medicina-59-01694-f002:**
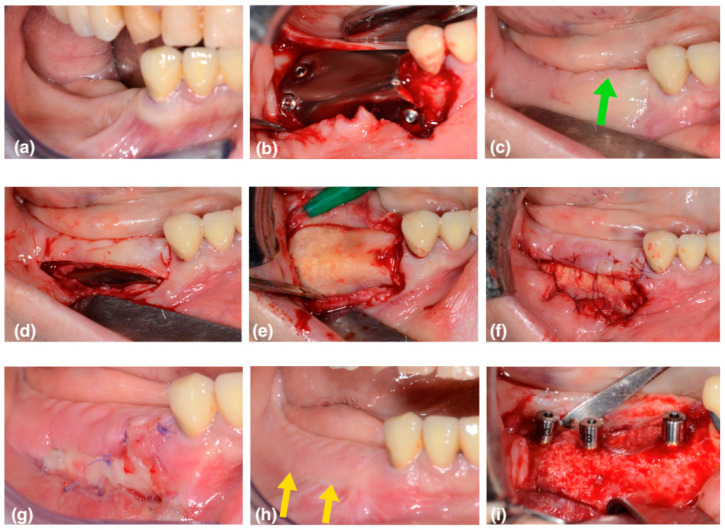
(**a**) Initial situation, it is evident the bone atrophy of the jaw. (**b**) First surgical phase of vertical bone augmentation with particulate graft and occlusive titanium barrier (OTB), fixed with three self-tapping screws. (**c**) The picture shows the soft tissue situation six months later, before the barrier removal. The green arrow highlights the coronal shifting of the mucogingival junction as result of the passivation of the flaps. (**d**) The buccal full-thickness incision is placed in the alveolar mucosa for the barrier removal. (**e**) A white layer of dense connective tissue, called the pseudoperiosteum, is observed underneath the OTB, protecting the newly formed hard tissue. (**f**) The buccal alveolar mucosa is apically sutured leaving intentionally exposed a vestibular portion of white pseudoperiosteum. (**g**) The picture shows the healing after 1 week. (**h**) The situation three months after the “white layer approach” (WLA) shows the new position of the mucogingival line as highlighted by the yellow arrows. (**i**) Implant placement in mature bone. It’s evident the difference in maturation of the bone compared to the picture (**e**).

**Figure 3 medicina-59-01694-f003:**
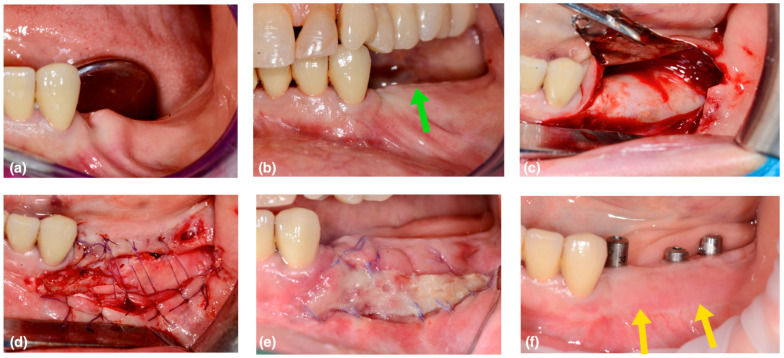
(**a**) Initial situation, it is evident the bone atrophy and the lack of keratinized tissue. (**b**) the picture shows the soft tissue situation 6 months after the vertical bone augmentation and before the barrier removal. The green arrow shows the coronal shifting of the mucogingival junction. (**c**) The picture shows the buccal incision placed in the alveolar mucosa and the barrier removal. (**d**) The buccal alveolar mucosa is apically sutured leaving intentionally exposed a vestibular portion of pseudoperiosteum. (**e**) The picture shows the healing after 1 week. (**f**) The situation after the “white layer approach” (WLA) and after the implant placement shows the new position of the mucogingival line as highlighted by the yellow arrows.

**Figure 4 medicina-59-01694-f004:**
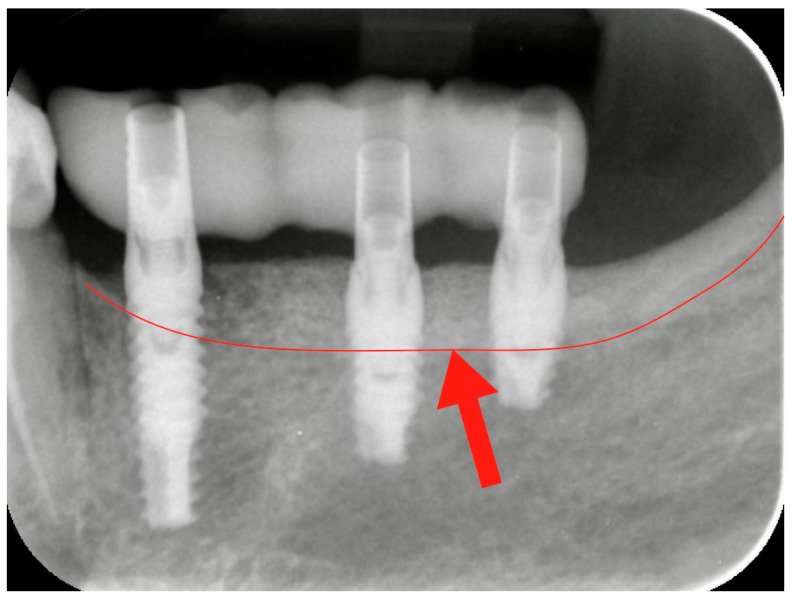
Radiographic check-up after the provisional resin restoration placement. The red line and the red arrow show the bone level before the vertical bone augmentation. The bone portion above the red line represents the amount of the vertical regeneration.

**Table 1 medicina-59-01694-t001:** The parameters evaluated in the study.

Patients	Sex	Implant Site	Age (Years)	VBG (mm)	ASTG (mm)
1	F	44-46-47	62	4	7
2	F	34-36-37	62	6	7.5
3	M	34-36	54	3.5	6
4	F	14-15-16	58	4	8
5	F	45-46	58	3	5
6	F	35-36	60	4	7
Average	-	-	59	4.08	6.75

## Data Availability

The data presented in this study are available on request from the corresponding author. The data are not publicly available due to privacy restrictions.

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
