# Peer review of "The “White Layer Approach”: A Graftless Gingival Augmentation Technique following Vertical GBR with Occlusive Titanium Barriers"

_medicina, 2023, doi:10.3390/medicina59101694_

Round 1

Reviewer 1 Report

1. What was the study design, prospective or retrospective? Although this is just a case series reporting 6 cases, it would be interesting to know if a uniform study design was made and IRB approval was obtained for the same.

2. A brief note in the introduction about the histological characteristics of the pseudo-periosteum would be of benefit to the readers.

3. After the 2nd phase surgery, it is seen that the pseudo periosteum layer was left exposed to the oral environment. Why was no attempt taken to cover it with perio-pack or any other suitable covering medium?

4. Which anatomical area was the source of autologous bone during 1st phase surgery? Was it the same for all patients?

Since this is a report of cases done, the methodology should be reported in past tense. Please correct the same along with other grammatical errors.

Author Response

  1. What was the study design, prospective or retrospective? Although this is just a case series reporting 6 cases, it would be interesting to know if a uniform study design was made and IRB approval was obtained for the same.
  • the study design was uniform (see materials and methods) and prospective, as the patient's response to the surgical procedure were followed from the beginning.
  • Institutional Review Board Statement: The study was performed in compliance with good clinical practice and the WMA Declaration of Helsinki for ethical principles in medical research involving human subjects (amended by the 64th WMA General Assembly, Fortaleza, Brazil, October 2013). Since a simple data collection was performed before and during the surgeries, the presence of an ethics committee was not required. 

2. A brief note in the introduction about the histological characteristics of the pseudo-periosteum would be of benefit to the readers.

ok, we will insert it.

3. After the 2nd phase surgery, it is seen that the pseudo periosteum layer was left exposed to the oral environment. Why was no attempt taken to cover it with perio-pack or any other suitable covering medium?

No perio-pack has been inserted over the pseudo-periostuem, because it can hinder the action of chlorhexidine during the healing phase (we will insert in the text). Furthermore, in our opinion, resorbable materials and connective subtitutes in this phase may increase the inflammatory response in the treated site, consequently leading to the possible resorption of the newly formed bone.

4. Which anatomical area was the source of autologous bone during 1st phase surgery? Was it the same for all patients? 

In the mandibular areas, the bone was harvested  from the retromolar area, on the external oblique line. In the upper cases, the bone harvest was performed from the zygomatic pillar. We will insert this part in the Methods, thank you.

Since this is a report of cases done, the methodology should be reported in past tense. Please correct the same along with other grammatical errors.

ok, we will insert it.

Reviewer 2 Report

The article presents an interesting method of gaining attached gingiva in fewer steps and with less morbidity than conventional surgery in cases requiring vertical bone augmentation and implant placement. The strength of the article is not the scientific research or statistical results, since there are only a few cases included, but the surgical aspect is of great value. Probably the small number of included cases is due to the novelty of the technique and it should be mentioned in the text.

The abstract should be modified since it does not summarize very well the content of the manuscript. The methodology presented in the abstract is not very clear regarding what happened at 6 months and at 3 months.

The methods section should mention only the parameters that were measured, the results of which should be presented in the results section. Therefore Table 1 should be in the results section. The results section should also include the surgery related outcomes. 

The number of patients and average age are presented both in the methods and results sections and probably should only be in the methods, representing the study group.

The English is not very easy to understand at times and the grammar not always accurate. 

Author Response

Good morning, thank you for the comments, you gave us some interesting comments to correct our article, and we will be able to provide the corrections as soon as possible. Below are the responses to your comments.

The article presents an interesting method of gaining attached gingiva in fewer steps and with less morbidity than conventional surgery in cases requiring vertical bone augmentation and implant placement. The strength of the article is not the scientific research or statistical results, since there are only a few cases included, but the surgical aspect is of great value. Probably the small number of included cases is due to the novelty of the technique and it should be mentioned in the text.

Ok, thank you, it'true, the small number is due to the novelty of the technique, we will mention this in the text, and we hope for more cases in future articles.

The abstract should be modified since it does not summarize very well the content of the manuscript. The methodology presented in the abstract is not very clear regarding what happened at 6 months and at 3 months.

Thank you, we will correct this aspect summarizing more clearly the methodology of the study in the abstract

The methods section should mention only the parameters that were measured, the results of which should be presented in the results section. Therefore Table 1 should be in the results section. The results section should also include the surgery related outcomes. 

Ok, thank you, we will correct this aspect in the article

The number of patients and average age are presented both in the methods and results sections and probably should only be in the methods, representing the study group.

Ok, thank you, we will correct and insert this data only in the methods